# Cardiac catheterization real-time dynamic radiation dose measurement to estimate lifetime attributable risk of cancer

Chun-Yuan Tu[1,2], Chung-Jung Lin[3,4], Bang-Hung Yang[1,5], Jay Wu[1], Tung-Hsin Wu[1]*

1 Department of Biomedical Imaging and Radiological Sciences, National Yang Ming University, Taipei, Taiwan, 2 Department of Radiology, Mackay Memorial Hospital, Taipei, Taiwan, 3 School of Medicine, National Yang-Ming University, Taipei, Taiwan, 4 Department of Radiology, Taipei Veterans General Hospital, Taipei, Taiwan, 5 Department of Nuclear Medicine, Taipei Veterans General Hospital, Taipei, Taiwan

* tung@ym.edu.tw

## Abstract

Cardiac catheterization procedure is the gold standard to diagnose and treat cardiovascular disease. However, radiation safety and cancer risk remain major concerns. This study aimed to real-time dynamic radiation dose measurement to estimate lifetime attributable risk (LAR) of cancer incidence and mortality in operators. Coronary angiography (CA) with percutaneous coronary intervention (PCI), CA, and others (radiofrequency ablation, pacemaker and defibrillator implantation) procedures with different beam directions, were undertaken on x-ray angiography system. A real-time electronic personal dosimeter (EPD) system was used to measure the radiation dose of staff during all procedures. We followed the Biological Effects of Ionizing Radiation (BEIR) VII report to estimate the LAR of all cancer incidence and mortality. Primary operators received radiation dose in CA with PCI, CA, and others procedures were 59.33 ± 95.03 µSv, 39.81 ± 103.85 µSv, and 21.92 ± 37.04 µSv, respectively. As to the assistant operators were 30.03 ± 55.67 µSv, 14.67 ± 14.88 µSv, and 4 µSv, respectively. LAR of all cancer incidences for staffs aged from 18 to 65 are varied from 0.40% for males to 1.50% for females. LAR of all cancer mortality for staffs aged from 18 to 65 are varied from 0.22% for males to 0.83% for females. Our study provided an easy, real-time and dynamic radiation dose measurement to estimate LAR of cancer for staff during the cardiac catheterization procedures. The LAR for all cancer incidence is about twice that for cancer mortality. Although the radiation doses of staff are lower during each procedure, the increased years of service leads to greater radiation risk to the staff.

## Introduction

Cardiac catheterization is an ionizing radiation procedure used to diagnose heart conditions or treat cardiovascular diseases. The procedures are well recognized to facilitate early and accurate diagnosis of the disease, improve treatment planning to save patient's life [1]. Nevertheless, the procedures usually perform with longer fluoroscopy time and may cause radiation

**Data Availability Statement:** All relevant data are within the manuscript and its Supporting Information files.

**Funding:** The author(s) received no specific funding for this work.

**Competing interests:** The authors have declared that no competing interests exist.

**Abbreviations:** LAR, lifetime attributable risk; CA, coronary angiography; PCI, percutaneous coronary intervention; EPD, electronic personal dosimeter; BEIR, biological effects of ionizing radiation.

exposure to staff [2–5]. Due to the correlation between exposure to ionizing radiation and cancer risk is related [6–11], staff are becoming increasingly aware of the potential damaging effects of ionizing radiation during the procedures. Thus, estimation of radiation dose and cancer risk in staff during the procedures is a major issue regarding the public health significance.

Although a number of studies had examined the radiation exposure during the interventional procedures in recent years, most of studies were performed on a phantom to simulate the radiation exposure to staff [12–17]. These phantom studies did not evaluate the dynamic changes in staff positions, beam orientation and movement, exposure parameters, and so on; moreover, in clinical procedures are often complex. Accordingly, the purpose of this study was real-time dynamics measurement of medical radiation dose to estimate the lifetime attributable risk (LAR) of cancer incidence and mortality in staff.

## Materials and methods

### Study design

The study was approved by the Mackay Memorial Hospital Institutional Review Board on June 22, 2017 and valid till June 21, 2018 (approval number: 17MMHIS075e). The constitution and operation of this review board are according to the guidelines of ICH-GCP, the records/information were anonymized and de-identified prior to analysis. All procedures were Data were collected for 3 different types of procedures: Coronary angiography (CA) with percutaneous coronary intervention (PCI), CA, and others (radiofrequency ablation, pacemaker and defibrillator implantation). Procedural details including types of procedure, fluoroscopy time, fluoroscopy tube voltage, fluoroscopy tube current, cine acquisition tube voltage, cine acquisition tube current, cine acquisition time, cine acquisition runs, dose area product (DAP), beam directions, staff (primary and assistant operator) age and radiation dose were recorded.

### Radiation dose measurement

Experimental measurements were used three x-ray angiography systems(one was Philips Allura FD20, the others were Philips Allura FD 10) with similar cardiac catheterization protocols. All protocols followed standard technical characteristics of image acquisition and quality control. Collimation and magnification were used during the procedures according to the clinical requirements. The operational protocols evaluated were fluoroscopy (15 pulses/s and 0.9 mm Cu as additional filtration) and cine acquisition (15 pulses/s without Cu filtration) modes. All staff adhered to standard radiation protection procedures. Each staff wore a lead apron, a thyroid collar, and leaded glasses. Because the thyroid is known to be radiosensitive and makes a significant contribution to the radiation dose [18–21], a real-time electronic personal dosimeter (EPD) system (i2, Raysafe) was placed over the left side of staff's thyroid collar to measure the radiation dose at various locations. EPD system has store instantaneous dose rate and cumulative dose values at the beginning to the end of each procedure. In addition, the system is design to measure the personal dose equivalent at depth of 10 mm (Hp(10)) for x-ray, and is considered to be the dose to the whole body [22].

### Cancer risk estimation

Currently, the linear no-threshold model is widely used to estimate the LAR of cancer from exposure to low levels of ionizing radiation. The LAR of cancer incidence and mortality, which are defined as additional cancer risk above and beyond baseline cancer risk. In this study, the LAR of all cancer incidence and mortality were calculated based on the Biological Effects of

Ionizing Radiation (BEIR) VII report [9]. Average radiation doses for each of the 3 procedure types were used to estimate the LAR of all cancer incidence and mortality. The LAR of all cancer incidence and mortality were estimated as follows:

$$\text{LAR(e, s, D)} = \int_{e+L}^{a\,Max} \text{ERR(e, s, D, a)}\, m\,(s, a)\,\frac{S(s, a)}{S(s, e)}\,da$$

Where (e) is the age at exposure, (s) is the sex specific excess relative risk (ERR), (D) is the dose of radiation received, (a) is the specific cancer site at attained age, the summation is from a = e + L to l00, where a denotes attained age (years) and L is a risk-free latent period (L = 5 for solid cancers; L = 2 for leukemia), ERR (e, s, D, a) is the risk model in the equation, m (s, a) is the baseline risk, S(s, a) is the probability of surviving until age (a), S(s, a)/S(s, e) is the probability of surviving to age (a) conditional on survival to age (e). This study estimated the cancer risk under the assumption that the operators were continuously exposed to radiation from the age of 18 to 65.

## Statistical analysis

Beam directions distribution of cardiac catheterization procedures was presented as percentage. Procedural details were presented as means and standard deviations by descriptive analyses. Multiple linear regression analysis was performed to DAP versus staff radiation dose. The *p* value < 0.05 was considered significant. LAR of cancer incidence and mortality for staff were expressed as line graphs.

## Results

There were 71 procedures were included in our study, 43 CA with PCI, 16 CA and 12 others. The beam directions distribution in CA with PCI, CA, and others procedures are illustrated in Fig 1. In CA with PCI procedure, the beam directions distribution of fluoroscopy and acquisition were most complexity. The procedural details in CA with PCI, CA, and others are listed in Table 1. Primary operators doses were measured under 43 CA with PCI, 16 CA, and 12 others procedures, respectively. As to the assistant operators doses were 38 CA with PCI, 6 CA, and 1 others procedures, respectively. Primary operators radiation dose in CA with PCI, CA, and others procedures were 59.33 ± 95.03 µSv, 39.81 ± 103.85 µSv, and 21.92 ± 37.04 µSv, respectively. As to the assistant operators were 30.03 ± 55.67 µSv, 14.67 ± 14.88 µSv, and 4 µSv, respectively. The fluoroscopy tube voltage of CA with PCI procedure (98.54±16.55 kV) was significantly higher than other two procedures, while the acquisition tube voltage was the same situation. The fluoroscopy time was longest in CA with PCI procedure (14.67±12.83 mins), followed by others procedure (14.21±12.73 mins) and CA procedure (6.10±3.49 mins). The acquisition time was also longest in CA with PCI procedure (53.16 ± 10.33 s), followed by CA procedure and others procedure. However, the fluoroscopy and acquisition tube current in others procedure were significantly lower than in the other two procedures. Correlation between DAP and staff radiation dose from all procedures are illustrated in Fig 2. Scatter graph of DAP versus primary operator (p = 0.004, $R^2$ = 0.11) and assistant operator radiation dose (p < 0.001, $R^2$ = 0.46) demonstrated weak positive correlations.

We used a 12 month average for the preceding year to estimate the 1 year occupational radiation exposure to the operators is presented in Fig 3. The annual average radiation dose per primary operator from all procedures was 3.49 mSv. As to the assistant operator was 1.30 mSv. Estimated LAR of all cancer incidence and mortality from cardiac catheterization procedures for operators are presented in Table 2. LAR of all cancer incidence and mortality for male primary operators aged from 18 to 65 were 1.07%, and 0.59%, respectively. As to

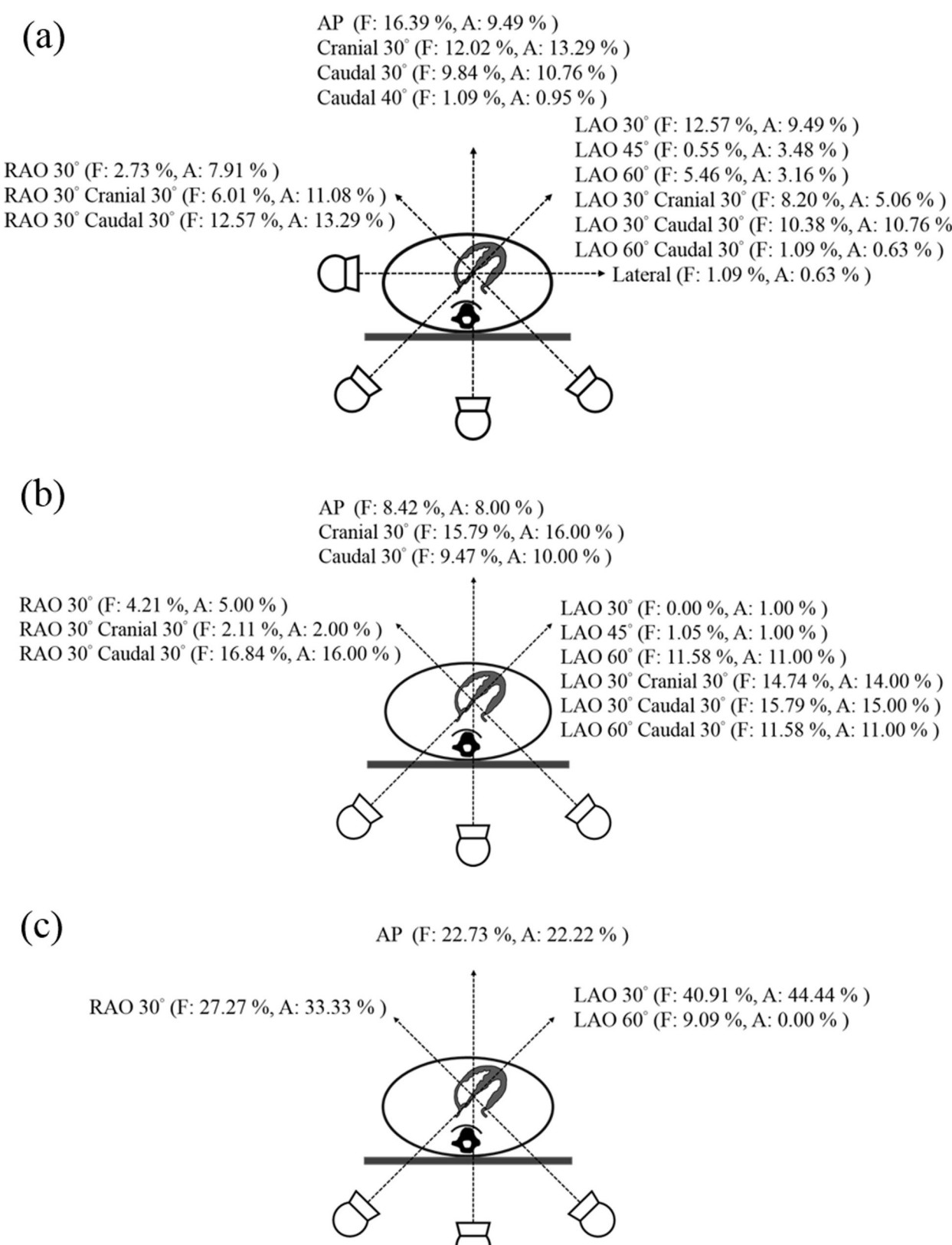

**Fig 1.** Beam directions distribution of cardiac catheterization procedures in (a) CA with PCI, (b) CA, and (c) Others. (LAO: left anterior oblique, RAO: right anterior oblique, AP: anterior posterior, CRAN: cranial, CAU: caudal, F: fluoroscopy, A: acquisition).

**Table 1. The procedural details in CA with PCI, CA, and others.**

|  | CA with PCI | CA | Others |
|---|---|---|---|
| Procedural details |  |  |  |
| Number of procedures (n = 71) | 43 | 16 | 12 |
| Radial approach (%) | 100 | 100 | 0 |
| Fluoroscopy time (mins) | 14.67 ± 12.83 | 6.10 ± 3.49 | 14.21 ± 12.73 |
| Fluoroscopy tube voltage (kV) | 98.54 ± 16.55 | 87.23 ± 15.06 | 93.61 ± 15.31 |
| Fluoroscopy tube current (mA) | 12.31 ± 4.93 | 15.62 ± 4.21 | 6.12 ± 2.93 |
| Acquisition tube voltage (kV) | 85.60 ± 14.94 | 80.84 ± 13.91 | 83.40 ± 20.61 |
| Acquisition tube current (mA) | 784.75 ± 107.18 | 733.46 ± 157.42 | 505.70 ± 300.73 |
| Acquisition time (s) | 53.16 ± 10.33 | 41.31 ± 11.56 | 4.42 ± 5.23 |
| Dose area product (Gy-cm$^2$) | 238.67 ± 201.51 | 119.30 ± 54.40 | 49.82 ± 126.27 |
| Primary operators (n = 6) |  |  |  |
| Number of primary operators | 5 | 4 | 3 |
| Age (years) | 49.25 ± 6.99 | 43.75 ± 3.30 | 42.33 ± 2.08 |
| Male (%) | 100 | 100 | 100 |
| Case volumes | 43 | 16 | 12 |
| EPD radiation dose (μSv) | 59.33 ± 95.03 | 39.81 ± 103.85 | 21.92 ± 37.04 |
| Assistant operators (n = 5) |  |  |  |
| Number of assistant operators | 5 | 3 | 1 |
| Age (years) | 32.6 ± 3.58 | 33.00 ± 4.58 | 32 |
| Male (%) | 44.74 | 83.33 | 100 |
| Case volumes | 38 | 6 | 1 |
| EPD radiation dose (μSv) | 30.03 ± 55.67 | 14.67 ± 14.88 | 4 |

the female primary operators aged from 18 to 65 were 1.50%, and 0.83%, respectively. In contrast, the LAR of all cancer incidence and mortality were significantly lower in assistant operators. The LAR of all cancer incidence and mortality for females were significantly higher than for males.

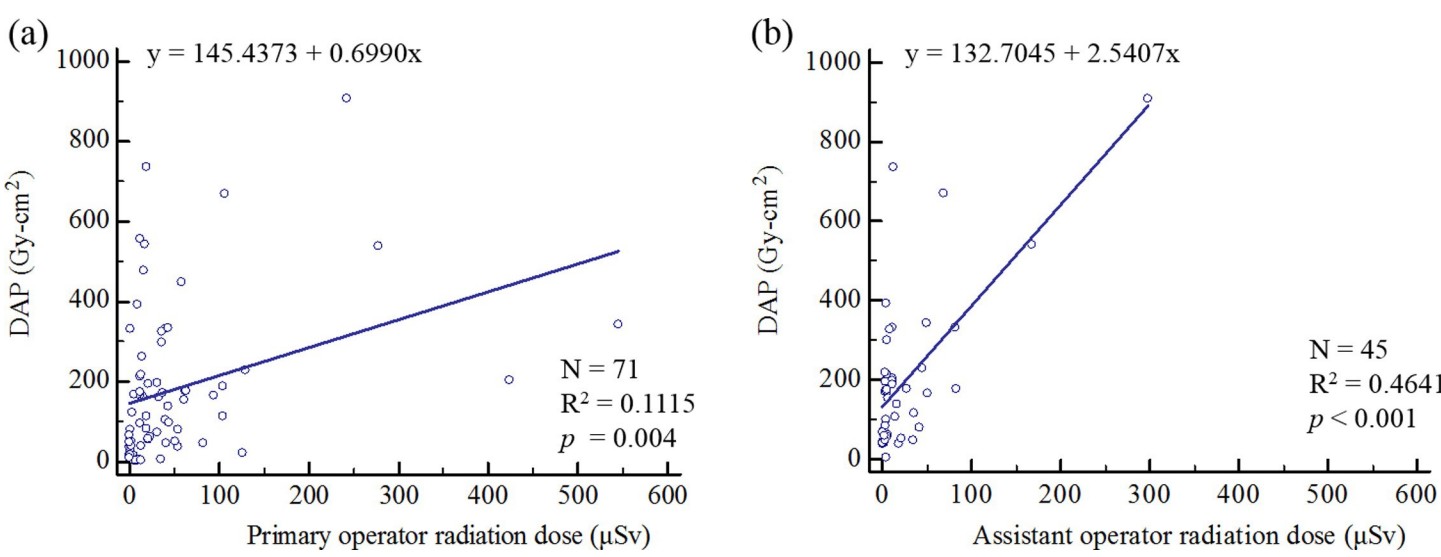

**Fig 2. Correlation of DAP versus staff radiation dose.** (a) DAP versus primary operator. (b) DAP versus assistant operator.

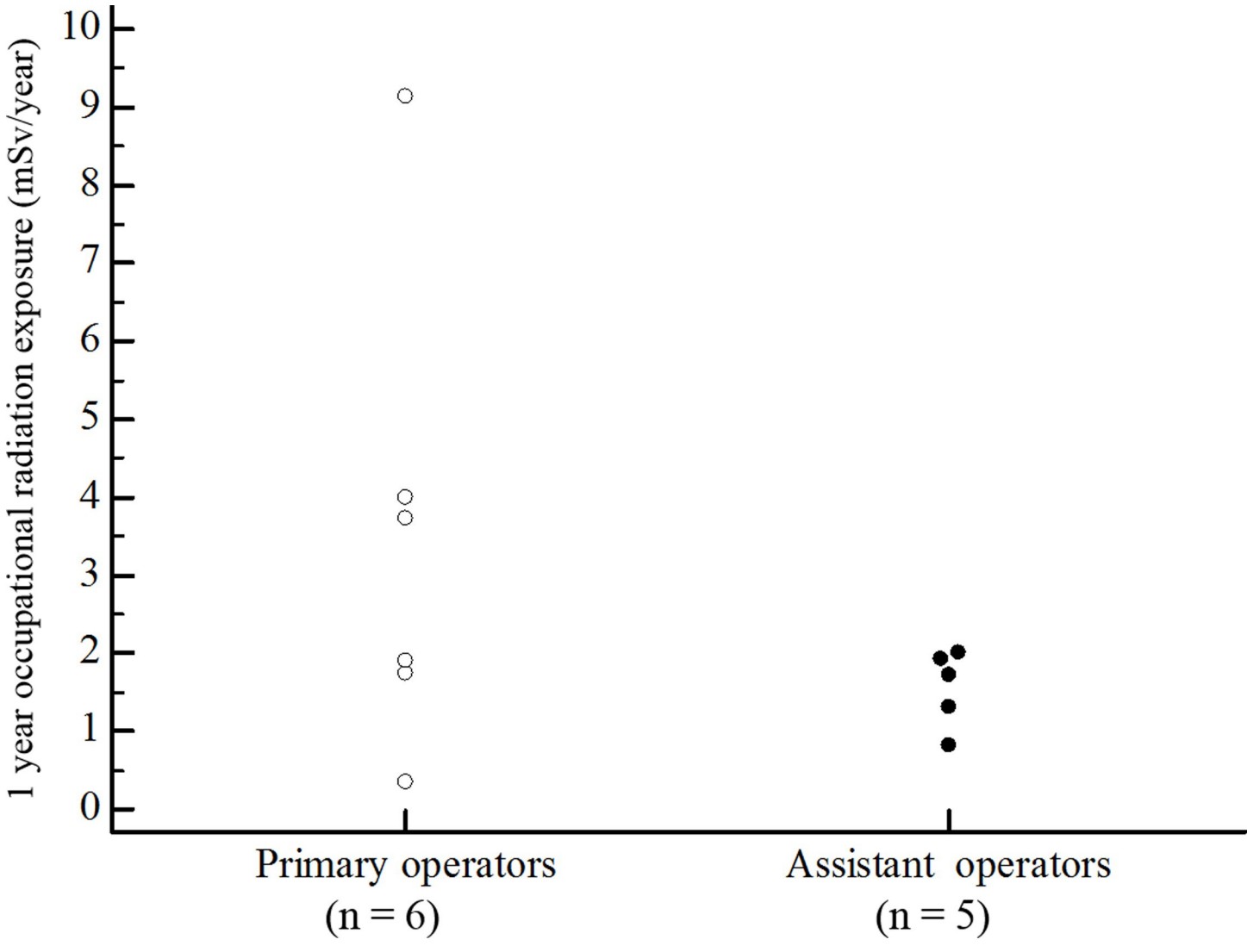

**Fig 3. The 1 year occupational radiation exposure to the operators.**

## Discussion

Many previous important phantom studies for similar cardiac catheterization procedures are listed in Table 3. However, our study was real-time dynamics measurement of medical radiation to estimate the link between medical radiation exposure and LAR of cancer in staff from cardiac catheterization procedures. Indeed, we analyzed the beam directions distribution complexity during the different procedures in this study. We have demonstrated the more complex

**Table 2. Estimated LAR of all cancer incidence and mortality from cardiac catheterization procedures for operators.**

| Operators | Primary operators | | Assistant operators | |
|---|---|---|---|---|
| LAR | All cancer incidence aged from 18 to 65 | All cancer mortality aged from 18 to 65 | All cancer incidence aged from 18 to 65 | All cancer mortality aged from 18 to 65 |
| Male | 1.07% | 0.59% | 0.40% | 0.22% |
| Female | 1.50% | 0.83% | 0.56% | 0.31% |

**Table 3. Previous phantom studies for similar cardiac catheterization procedures.**

| Author | Location of dosimeter | Measurement tool | Dose unit | No of projections |
|---|---|---|---|---|
| Panetta et al.[32] | Wrist | EPD | Dose rate | 4 |
| Patet et al.[33] | Chest | EPD | Equivalent dose | _ |
| Etzel et al.[34] | Eye/ Neck/ Chest/ Gonads/ Lower leg | Ion chamber | Dose rate | 3 |
| Jia et al.[35] | Eye/ Neck/ Chest/ Epigastrium/ Hypogastrium/ Thigh/ Lower leg/ Ankle | EPD | Dose rate | 8 |
| Haga et al.[36] | Eye/ Neck | EPD/ Eye dosimeter | Equivalent dose | - |
| Alnewaini et al.[37] | Eye/ Neck | TLD | Radiation dose | 14 |
| Oliveira da Silva et al.[38] | Eye/ Chest | EPD | Equivalent dose | 6 |
| Perisinakis et al.[39] | Eye/ Waist | Ion chamber | Dose rate | 17 |
| Ordiales et al.[15] | Neck | EPD | Equivalent dose | 7 |
| Sciahbasi et al.[40] | Head/ Chest/ Wrist/ Hip | EPD | Equivalent dose | 8 |
| Vano et al.[41] | Chest/ Eye | EPD | Dose rate | 13 |
| Principi et al.[42] | Neck/ Chest/ Shoulder | EPD/ TLD | Equivalent dose | 2 |
| Liu et al.[43] | Chest | TLD | Effective dose | 6 |
| Farah et al.[44] | Eye/Neck/Chest/Waist | EPD/ TLD | Equivalent dose | 10 |
| Ertel et al.[45] | Chest | Ion chamber | Radiation dose | 7 |
| Chida et al.[46] | Neck/ Chest/ Knee | EPD | Dose rate | - |
| Boetticher et al.[47] | Eye/ Neck/ Chest/ Gonads/ Knee/ Lower leg/ Foot | TLD | Effective dose | 3 |
| Mesbahi et al.[48] | From head to foot (for every 10 cm) | Ion chamber | Dose rate | - |
| Schultz et al.[49] | Trunk | EPD | Effective dose | 2 |
| Koichi et al.[50] | Neck/ Chest | OSLD* | Dose rate | 4 |
| Kuon et al.[51] | Chest | Ion chamber | Dose rate | 163 |
| Balter et al.[52] | Neck/ Chest/ Knee | Ion chamber | Dose rate | 6 |

*OSLD: optically stimulated luminescence dosimeter.

procedures are associated with increasing the radiation doses. Estimation of the radiation doses in staff have a wide variation across the literatures as a result of the levels of complexity of the procedures [1, 3, 23–25]. This phenomenon is comparable with our staff radiation doses. Nevertheless, the assistant operators radiation doses were significantly lower than primary operators is observed in all procedures. This result is caused by many factors during the procedures such as equipment set-up, operator technique, use of radiation reducing techniques, workload and procedural complexity [26–29].

There have been many studies[1, 30, 31] to estimate effective dose or effective dose equivalent using personal monitors. The information from these studies could be used in evaluating likely dose levels. The modified Niklason algorithm provided a measure of the exposure of sensitive organs in the trunk:

$$E = 0.02 \, (H_{OS} - H_{U}) + \, H_{U} \text{ (with a thyroid collar)}$$

where E is effective dose, Hos is Hp (0.07) measured over shield on thyroid level, and Hu is Hp (10) measured under apron. The Hp (10) over shield on thyroid level is converted to Hp (0.07) by adding 3% to the measurement dose. For a single dosimeter worn at the thyroid collar, again assuming $H_{U} \approx 0.01$ Hos, the conversion algorithm as follows:

$$E = 0.03 \, H_{OS}$$

Martin et al.[31] is recommended for estimating the eye dose from a measurement with an unshielded neck dosimeter, the equation:

$$\text{Eye dose} = 0.75 \times \text{neck dose}$$

Cardiac catheterization procedures provide great diagnostic and treatment benefit to patients. Unfortunately, the radiation doses of these procedures are imposed on staff. The radiation dose to staff during the procedure is due to Compton scatter in the patient which is the dominant interaction in tissue at diagnostic x-ray energies [28, 53]. Modern angiography system provides DAP values to monitor the radiation dose for patient during the procedures. From some literatures reported a positive correlation between DAP and staff radiation dose during the procedures [22, 29, 54]. This means that DAP values can be represented the relative radiation dose of the staff. Nevertheless, our results demonstrated a weak correlation. This could be due to several factors: first, the quantity of radiation varies significantly depending on the position of staff relative to the x-ray source and patient; second, the staff used a ceiling-mounted radiation shielding screen for radiation protection; third, the staff might leave the cardiac catheterization lab during the acquisition. This implies that the radiation dose to staff during the procedures might be reduced by improving radiation protection practices.

Although our study demonstrates that the radiation doses of staff are lower during each procedure, the increased years of service leads to greater radiation risk to the operators. In fact, the radiation risks are mainly stochastic effects at low radiation exposure levels. These effects with the probability of occurrence increasing with the absorbed dose [28, 55]. Therefore, lead aprons and thyroidal collars and leaded glasses allow staff to achieve As Low As Reasonably Achievable (ALARA) during the procedures. As was shown in an earlier report [28], lead aprons of 0.25 or 0.5 mm absorb 85–92% 100 kV of energy and 93–97% of 100 kV of energy, respectively. All operators wear lead aprons, thyroid collars, and leaded glasses to protect themselves, during long-term performance of the procedures, it is still impossible to completely avoid radiation exposure and its effects. If operators fail to use protective gear or adjust the exposure time properly, within a few years operators may have increased the LAR of cancer. Optimizing the fluoroscopy or cine acquisition dose rate to reduce staff radiation dose and long-term risk is much more efficient. Ishibashi et al. [56] reported the fluoroscopy dose rates in Japan were 7.5 pulses/s at 44% of CA, 7.5 pulses/s at 43% of PCI, and 7.5 pulses/s at 54% of radiofrequency catheter ablationpulses, respectively. As to the cine acquisition dose rates were 15 pulses/s at 93% of CA, 15 pulses/s at 90% of PCI, and 7.5 pulses/s at 55% of radiofrequency catheter ablationpulses, respectively. van Dijk et al. [57] reported the fluoroscopy dose rate in pacemaker and defibrillator implantation was 7.5 pulses/s. As to the cine acquisition dose rates were 3.75–15 pulses/s. In our study, the default pulse rates (15 pulses/s) both fluoroscopy and cine acquisition dose rates were used in other procedures. Consequently, the fluoroscopy and cine acquisition dose rates would be optimized to ensure that the radiation dose would be reduced to an acceptable level.

There are a few limitations in our study. First of all, due to the characteristics of equipment and method of radiation dose measurement which lead to each beam direction and field size variation were not straightforward determined during the procedure. Our study did not analyze the correlation between each beam direction, field size variation and radiation dose. Second, the presented exposure levels reflected the procedure types were most often performed at two hospitals. This certainly did not cover the all types of procedures and which was affected the sample size. Third, the hand doses may be much greater than doses at the neck, eye, or trunk during the procedures. There is no conversion algorithm to estimate the hand dose from the doses at neck. The dosimeter should be worn towards on the hand adjacent to the x-ray

tube if a meaningful result is to be obtained. Fourth, BEIR VII report was based on a linear no-threshold (LNT) model to assess the correlation between radiation exposure and biological risk. The principal source was the effect of whole body acute exposure to high radiation dose. Whether the principal source can be extrapolated to the partial body exposure at a much lower radiation dose. In addition, our study did not estimate the yearly of radiation exposure exactly and job tenure of operators. However, according to the BEIR VII report, when exposed continuously to 10 mSv on a yearly basis from ages 18 to 65 years old, cancer incidence was 3,059 for male and 4,295 for female. That data was directly applied to the calculation of cancer risk to medical staff in many studies [10, 58–60]. Although the LNT model for low-dose (<100mSv) is characterized by a great deal of uncertainty, there is no direct evidence to estimate the cancer risks for staff during the cardiac catheterization procedures. Currently BEIR VII report offers the most accurate estimates of cancer incidence and mortality from medical radiation dose.

## Conclusion

The radiation dose to staff mainly depend on the large number of acquisition and longer fluoroscopy time during the procedures. The present study provided an easy, real-time and dynamic radiation dose measurement to estimate LAR of cancer for staff in the cardiac catheterization procedures. Although the radiation doses of staff are lower during each procedure, the increased years of service leads to greater radiation risk to the staff. In addition, the LAR for all cancer incidence is about twice that for cancer mortality. Given the limits of the correlation between each beam direction, field size variation and staff radiation dose of this study. In our future studies will be of value to validate these findings.

## Supporting information

**S1 Data.**
(XLSX)

## Author Contributions

**Conceptualization:** Chun-Yuan Tu, Chung-Jung Lin, Bang-Hung Yang, Tung-Hsin Wu.

**Data curation:** Chun-Yuan Tu, Chung-Jung Lin, Bang-Hung Yang, Tung-Hsin Wu.

**Formal analysis:** Chun-Yuan Tu, Tung-Hsin Wu.

**Methodology:** Chun-Yuan Tu, Jay Wu, Tung-Hsin Wu.

**Resources:** Chun-Yuan Tu.

**Software:** Chun-Yuan Tu, Jay Wu.

**Validation:** Chun-Yuan Tu, Chung-Jung Lin, Bang-Hung Yang, Jay Wu.

**Writing – original draft:** Chun-Yuan Tu, Tung-Hsin Wu.

**Writing – review & editing:** Chung-Jung Lin, Bang-Hung Yang, Jay Wu, Tung-Hsin Wu.

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
