## [Decision Letter · Decision Letter 0]

6 Feb 2020

PONE-D-19-29699

Cardiac Catheterization Real-time Dynamic Radiation Dose Measurement to Estimate Lifetime Attributable Risk of Cancer

PLOS ONE

Dear Dr. Wu,

Thank you for submitting your manuscript to PLOS ONE. After careful consideration, we feel that it has merit but does not fully meet PLOS ONE’s publication criteria as it currently stands. Therefore, we invite you to submit a revised version of the manuscript that addresses the points raised during the review process.

Overall, the reviewers were interested in the manuscript, but had some methodological questions that will clarify some of the interpretations of the results. There were also some grammar errors that should be addressed for readability. 

We would appreciate receiving your revised manuscript by Mar 22 2020 11:59PM. To enhance the reproducibility of your results, we recommend that if applicable you deposit your laboratory protocols in protocols.io, where a protocol can be assigned its own identifier (DOI) such that it can be cited independently in the future. For instructions see: http://journals.plos.org/plosone/s/submission-guidelines#loc-laboratory-protocols

We look forward to receiving your revised manuscript.

Kind regards,

Jay Widmer

Academic Editor

PLOS ONE

Journal Requirements:

2. Please provide additional details regarding participant consent. In the ethics statement in the Methods and online submission information, please ensure that you have specified what type of consent you obtained (for instance, written or verbal). If you obtained verbal consent, please state why it was not possible to obtain written consent, how verbal consent was recorded and whether the ethics committee approved this consent procedure.

Reviewers' comments:

Reviewer's Responses to Questions

**Comments to the Author**

1. Is the manuscript technically sound, and do the data support the conclusions?

Reviewer #1: Partly

Reviewer #2: Partly

2. Has the statistical analysis been performed appropriately and rigorously? 

Reviewer #1: Yes

Reviewer #2: N/A

3. Have the authors made all data underlying the findings in their manuscript fully available?

Reviewer #1: No

Reviewer #2: Yes

4. Is the manuscript presented in an intelligible fashion and written in standard English?

Reviewer #1: Yes

Reviewer #2: No

5. Review Comments to the Author

Reviewer #1: 1. please provide a table (in discussion) listing all important phantom studies for similar cardiac procedures.

2. How many personnale were included for dosimetric measurements ? Please list the number of each staff-e.g. How many primary operators, how many assistant operators etc. How many women and what age ?

3. The EPD was worn around the collar, what about the chest, hands, ovaries or pelvis in women. Even if there were no EPD measurements in these body sites, an estimate should be given and discussed.

4. Line 231-232 is incomplete, please fix grammar throughout the document.

5. Its not clear if the EPD measurements listed are an average exposure of each individual or average of the entire group. There is a need for a major clarification- are the measured doses from one or an average of all personnel

6. How many such procedures are done in a year and how much cumulative exposure does a person get exposed to in a year? that estimate has to be provided.

7. How does the exposure differ between various machine models?

Reviewer #2: 1. Line 50 -- Cardiac catheterization should not be described as a "radiologic" procedure. Rather, it is ionizing radiation-based.

2. "other procedures" often use frame rates of 4/s or 7.5/s, considerably lower than the 15 fps listed here. Please acknowledge this in the Discussion.

3. The manuscript leaves very unclear what is being calculated. Is it the radiation dose per case, per year, or per career. This is crucial to the manuscript's conclusions. For example, calculating the LAR of a 60 year old interventional cardiologist based on one year's exposure isn't meaningful, since that individual has likely had exposures beginning 30 years previously. Complicating factors even further, exposure times were likely longer when the individual was younger and less experienced, and the yearly case load may have been lower as the individual was building a career and reputation. Exactly what is being calculated is critical here.

4. The first table should include the ages and case volumes of the operators under study.

5. How was the volume of 450 cases/year obtained?

6. How was continual and correct placement of the dosimeter assured?

7. Are approaches to PCI predominantly femoral or radial?

8. The beam angles in Fig 1 add up to more than 100%.

9. One assumes that he dose-area products recorded are those for the patient. Is this correct?

10. The fluoroscopy and acquisition times for PCI seem very short. Please double check.

11. How many procedures were used for the measurements?

12. Was there a specific protocol for edge-to edge collimation?

13. Lines 175-176 seem to be erroneous. This study didn't report relations between beam angles and radiation dose. Was this from a previous publication.

6. PLOS authors have the option to publish the peer review history of their article (what does this mean?). If published, this will include your full peer review and any attached files.

Reviewer #1: No

Reviewer #2: No

---

## [Author Response · Author response to Decision Letter 0]

20 Mar 2020

Dear Editor, Dear reviewers

We thank the reviewers for the helpful comments. They are all very important points and definitely improve the quality of this manuscript. The following is a point-by-point response to each comment:

Reviewer #1:

1. please provide a table (in discussion) listing all important phantom studies for similar cardiac procedures.

Author response:

Thank you for the instructions. We have rephrased the statement in line 171-174 as “Many previous important phantom studies for similar cardiac catheterization procedures are listed in Table 3. However, our study was real-time dynamics measurement of medical radiation to estimate the link between medical radiation exposure and LAR of cancer in staff from cardiac catheterization procedures.”

We have added a table in the main manuscript (in discussion) as follows:

Table 3 Previous phantom studies for similar cardiac catheterization procedures

Newly added references:

30. Padovani R, Rodella C. Staff dosimetry in interventional cardiology. Radiation protection dosimetry. 2001;94(1-2):99-103.

31. Martin C. Personal dosimetry for interventional operators: when and how should monitoring be done? The British journal of radiology. 2011;84(1003):639-48.

32. Panetta CJ, Galbraith EM, Yanavitski M, Koller PK, Shah B, Iqbal S, et al. Reduced radiation exposure in the cardiac catheterization laboratory with a novel vertical radiation shield. Catheterization and Cardiovascular Interventions. 2020;95(1):7-12.

33. Patet C, Ryckx N, Arroyo D, Cook S, Goy JJ. Efficacy of the SEPARPROCATH® radiation drape to reduce radiation exposure during cardiac catheterization: A pilot comparative study. Catheterization and Cardiovascular Interventions. 2019;94(3):387-91.

34. Etzel R, König AM, Keil B, Fiebich M, Mahnken AH. Effectiveness of a new radiation protection system in the interventional radiology setting. European journal of radiology. 2018;106:56-61.

35. Jia Q, Chen Z, Jiang X, Zhao Z, Huang M, Li J, et al. Operator Radiation and the Efficacy of Ceiling-Suspended Lead Screen Shielding during Coronary Angiography: An Anthropomorphic Phantom Study Using Real-Time Dosimeters. Scientific reports. 2017;7:42077.

36. Haga Y, Chida K, Kaga Y, Sota M, Meguro T, Zuguchi M. Occupational eye dose in interventional cardiology procedures. Scientific reports. 2017;7(1):1-7.

37. Alnewaini Z, Langer E, Schaber P, David M, Kretz D, Steil V, et al. Real‐time, ray casting‐based scatter dose estimation for c‐arm x‐ray system. Journal of applied clinical medical physics. 2017;18(2):144-53.

38. Oliveira da Silva M, Canevaro L, Hunt J, Rodrigues B. Comparing Measured and Calculated Doses in Interventional Cardiology Procedures. Radiation protection dosimetry. 2017;176(4):439-43.

39. Perisinakis K, Solomou G, Stratakis J, Damilakis J. Data and methods to assess occupational exposure to personnel involved in cardiac catheterization procedures. Physica Medica. 2016;32(2):386-92.

40. Sciahbasi A, Rigattieri S, Sarandrea A, Cera M, Di Russo C, Fedele S, et al. Operator radiation exposure during right or left transradial coronary angiography: a phantom study. Cardiovascular Revascularization Medicine. 2015;16(7):386-90.

41. Vano E, Sanchez R, Fernandez J, Bartal G, Canevaro L, Lykawka R, et al. A set of patient and staff dose data for validation of Monte Carlo calculations in interventional cardiology. Radiation protection dosimetry. 2015;165(1-4):235-9.

42. Principi S, Ginjaume M, Duch MA, Sánchez RM, Fernández JM, Vano E. Influence of dosemeter position for the assessment of eye lens dose during interventional cardiology. Radiation protection dosimetry. 2015;164(1-2):79-83.

43. Liu H, Jin Z, Jing L. Comparison of radiation dose to operator between transradial and transfemoral coronary angiography with optimised radiation protection: a phantom study. Radiation protection dosimetry. 2014;158(4):412-20.

44. Farah J, Struelens L, Dabin J, Koukorava C, Donadille L, Jacob S, et al. A correlation study of eye lens dose and personal dose equivalent for interventional cardiologists. Radiation protection dosimetry. 2013;157(4):561-9.

45. Ertel A, Nadelson J, Shroff AR, Sweis R, Ferrera D, Vidovich MI. Radiation Dose Reduction during Radial Cardiac Catheterization: Evaluation of a Dedicated Radial Angiography Absorption Shielding Drape. ISRN Cardiol. 2012;2012:769167. Epub 2012/09/19. doi: 10.5402/2012/769167. PubMed PMID: 22988525; PubMed Central PMCID: PMCPMC3439952.

46. Chida K, Morishima Y, Inaba Y, Taura M, Ebata A, Takeda K, et al. Physician-received scatter radiation with angiography systems used for interventional radiology: comparison among many X-ray systems. Radiation protection dosimetry. 2012;149(4):410-6.

47. von Boetticher H, Lachmund J, Hoffmann W. Cardiac catheterization: impact of face and neck shielding on new estimates of effective dose. Health Physics. 2009;97(6):622-7.

48. Mesbahi A, Mehnati P, Keshtkar A, Aslanabadi N. Comparison of radiation dose to patient and staff for two interventional cardiology units: a phantom study. Radiation protection dosimetry. 2008;131(3):399-403.

49. Schultz F, Zoetelief J. Dosemeter readings and effective dose to the cardiologist with protective clothing in a simulated interventional procedure. Radiation protection dosimetry. 2008;129(1-3):311-5.

2. How many personnale were included for dosimetric measurements ? Please list the number of each staff-e.g. How many primary operators, how many assistant operators etc. How many women and what age ?

Author response:

Thank you for the correction. We have added more specific details about the operators characteristics in the Table 1 as follows:

Table 1 The procedural details in CA with PCI, CA, and others

3. The EPD was worn around the collar, what about the chest, hands, ovaries or pelvis in women. Even if there were no EPD measurements in these body sites, an estimate should be given and discussed.

Author response:

Thank you for pointing this out. We have added more specific details about the radiation doses of the different body sites in discussion as follows:

We have rephrased the statement in line 185-198 as “There have been many studies[1, 30, 31] to estimate effective dose or effective dose equivalent using personal monitors. The information from these studies could be used in evaluating likely dose levels. The modified Niklason algorithm provided a measure of the exposure of sensitive organs in the trunk:

E = 0.02 (HOS − HU) + HU (with a thyroid collar)

where E is effective dose, Hos is Hp (0.07) measured over shield on thyroid level, and Hu is Hp (10) measured under apron. The Hp (10) over shield on thyroid level is converted to Hp (0.07) by adding 3% to the measurement dose. For a single dosimeter worn at the thyroid collar, again assuming HU ≈ 0.01 Hos, the conversion algorithm as follows:

E = 0.03 HOS 

Martin et al.[31] is recommended for estimating the eye dose from a measurement with an unshielded neck dosemeter, the equation: 

Eye dose = 0.75 × neck dose”

Newly added references:

30. Padovani R, Rodella C. Staff dosimetry in interventional cardiology. Radiation protection dosimetry. 2001;94(1-2):99-103.

31. Martin C. Personal dosimetry for interventional operators: when and how should monitoring be done? The British journal of radiology. 2011;84(1003):639-48.

We have rephrased the statement in line 192-205 as” Third, the hand doses may be much greater than doses at the neck, eye, or trunk during the procedures. There is no conversion algorithm to estimate the hand dose from the doses at neck. The dosimeter should be worn towards on the hand adjacent to the x-ray tube if a meaningful result is to be obtained.”

4. Line 231-232 is incomplete, please fix grammar throughout the document.

Author response:

Thank you for the correction. We have rephrased the statement in line 253-255 as “Although the LNT model for low-dose (<100mSv) is characterized by a great deal of uncertainty. As it for cardiac imagines, there is no direct evidence to estimate the cancer risks.” 

5. Its not clear if the EPD measurements listed are an average exposure of each individual or average of the entire group. There is a need for a major clarification- are the measured doses from one or an average of all personnel

Author response:

Thank you for your valuable opinions. We have rephrased the statement in line 130-133 as “Primary operators doses were measured under 43 CA with PCI, 16 CA, and 12 others procedures, respectively. As to the assistant operators doses were 38 CA with PCI, 6 CA, and 1 others procedures, respectively.”

We also have added more details about the case volumes of the operators under this study in the Table 1 as follows:

Table 1 The procedural details in CA with PCI, CA, and others

6. How many such procedures are done in a year and how much cumulative exposure does a person get exposed to in a year? that estimate has to be provided.

Author response:

Thank you for your valuable opinions. We have rephrased the statement in line 155-158 as “We used the data from 71 procedures within 2 months to estimate the 1 year occupational radiation exposure to the operators is presented in Fig. 3. The annual average radiation dose per primary operator from all procedures was 3.28 mSv. As to the assistant operator was 1.59 mSv.”

7. How does the exposure differ between various machine models?

Author response:

Thank you for the correction. We have added more details about the exposure differ between various machine models in materials and methods as follows:

We have rephrased the statement in line 83-87 as “Experimental measurements were used three x-ray angiography systems(one was Philips Allura FD20, the others were Philips Allura FD 10) with similar cardiac catheterization protocols. All protocols followed standard technical characteristics of image acquisition and quality control. Collimation and magnification were used during the procedures according to the clinical requirements.”

Reviewer #2: 

1. Line 50 -- Cardiac catheterization should not be described as a "radiologic" procedure. Rather, it is ionizing radiation-based.

Author response:

Thank you for your valuable opinions. We have rephrased the statement in line 50-51 as “Cardiac catheterization is an ionizing radiation procedure used to diagnose heart conditions or treat cardiovascular diseases.”

2. "other procedures" often use frame rates of 4/s or 7.5/s, considerably lower than the 15 fps listed here. Please acknowledge this in the Discussion.

Author response:

Thank you for pointing this out. We have added more details about the frame rates of other procedures in discussion as follows:

We have rephrased the statement in line 227-238 as “Optimizing the fluoroscopy or cine acquisition dose rate to reduce staff radiation dose and long-term risk is much more efficient. Ishibashi et al. [56] reported the fluoroscopy dose rates in Japan were 7.5 pulses/s at 44 % of CA, 7.5 pulses/s at 43 % of PCI, and 7.5 pulses/s at 54 % of radiofrequency catheter ablationpulses, respectively. As to the cine acquisition dose rates were 15 pulses/s at 93 % of CA, 15 pulses/s at 90 % of PCI, and 7.5 pulses/s at 55 % of radiofrequency catheter ablationpulses, respectively. van Dijk et al. [57] reported the fluoroscopy dose rate in pacemaker and defibrillator implantation was 7.5 pulses/s. As to the cine acquisition dose rates were 3.75-15 pulses/s. In our study, the default pulse rates (15 pulses/s) both fluoroscopy and cine acquisition dose rates were used in other procedures. Consequently, the fluoroscopy and cine acquisition dose rates must be optimized to ensure that the radiation dose be reduced to an acceptable level.”

Newly added references:

56. Ishibashi T, Takei Y, Sakamoto H, Yamashita Y, Kato M, Tsukamoto A, et al. Nationwide Survey of Medical Radiation Exposure on Cardiovascular Examinations in Japan. Nihon Hoshasen Gijutsu Gakkai zasshi. 2020;76(1):64.

57. van Dijk JD, Ottervanger JP, Delnoy PPH, Lagerweij MC, Knollema S, Slump CH, et al. Impact of new X-ray technology on patient dose in pacemaker and implantable cardioverter defibrillator (ICD) implantations. Journal of Interventional Cardiac Electrophysiology. 2017;48(1):105-10.

3. The manuscript leaves very unclear what is being calculated. Is it the radiation dose per case, per year, or per career. This is crucial to the manuscript's conclusions. For example, calculating the LAR of a 60 year old interventional cardiologist based on one year's exposure isn't meaningful, since that individual has likely had exposures beginning 30 years previously. Complicating factors even further, exposure times were likely longer when the individual was younger and less experienced, and the yearly case load may have been lower as the individual was building a career and reputation. Exactly what is being calculated is critical here.

Author response:

Thank you for pointing this out. We have added more details about the cancer risk estimation in abstract, materials and methods, results, discussion and conclusion as follows:

We have rephrased the statement in line 32-39 (abstract) as” LAR of all cancer incidences for staffs aged from 18 to 65 are varied from 0.49 % for males to 1.41 % for females. LAR of all cancer mortality for staffs aged from 18 to 65 are varied from 0.27 % for males to 0.78 % for females. Our study provided an easy, real-time and dynamic radiation dose measurement to estimate LAR of cancer for staff during the cardiac catheterization procedures. The LAR for all cancer incidence is about twice that for cancer mortality. Although the radiation doses of staff are lower during each procedure, the increased years of service leads to greater radiation risk to the staff.”

We have rephrased the statement in line 115-117 (materials and methods) as “This study estimated the cancer risk under the assumption that the operators were continuously exposed to radiation from the age of 18 to 65.”

We have rephrased the statement in line 158-165 (results) as “Estimated LAR of all cancer incidence and mortality from cardiac catheterization procedures for operators are presented in Table 2. LAR of all cancer incidence and mortality for male primary operators aged from 18 to 65 were 1.00 %, and 0.56 %, respectively. As to the female primary operators aged from 18 to 65 were 1.41 % and 0.78 %, respectively. In contrast, the LAR of all cancer incidence and mortality were significantly lower in assistant operators. The LAR of all cancer incidence and mortality for females were significantly higher than for males.”

Table 2 Estimated LAR of all cancer incidence and mortality from cardiac catheterization procedures for operators

We have rephrased the statement in line 215-227 (discussion) as “Although our study demonstrates that the radiation doses of staff are lower during each procedure, the increased years of service leads to greater radiation risk to the operators. In fact, the radiation risks are mainly stochastic effects at low radiation exposure levels. These effects with the probability of occurrence increasing with the absorbed dose [28, 55]. Therefore, lead aprons and thyroidal collars and leaded glasses allow staff to achieve As Low As Reasonably Achievable (ALARA) during the procedures. As was shown in an earlier report [28], lead aprons of 0.25 or 0.5 mm absorb 85-92 % 100 kV of energy and 93-97 % of 100 kV of energy, respectively. All operators wear lead aprons, thyroid collars, and leaded glasses to protect themselves, during long-term performance of the procedures, it is still impossible to completely avoid radiation exposure and its effects. If operators fail to use protective gear or adjust the exposure time properly, within a few years operators may have increased the LAR of cancer.”

We have rephrased the statement in line 261-265 (conclusion) as “The present study provided an easy, real-time and dynamic radiation dose measurement to estimate LAR of cancer for staff in the cardiac catheterization procedures. Although the radiation doses of staff are lower during each procedure, the increased years of service leads to greater radiation risk to the staff.”

4. The first table should include the ages and case volumes of the operators under study.

Author response:

Thank you for your valuable opinions. We have added more details about the ages and case volumes of the operators under this study in the Table 1 as follows:

Table 1 The procedural details in CA with PCI, CA, and others

5. How was the volume of 450 cases/year obtained?

Author response:

Thank you for pointing this out. We have added more details about the cumulative exposure does a person get exposed to in a year as follows:

We have rephrased the statement in line 155-158 as “We used the data from 71 procedures within 2 months to estimate the 1 year occupational radiation exposure to the operators is presented in Fig. 3. The annual average radiation dose per primary operator from all procedures was 3.28 mSv. As to the assistant operator was 1.59 mSv.”

6. How was continual and correct placement of the dosimeter assured?

Author response:

Before the all measurements, the dosimeter was attached over the operator’s thyroid collar at a fixed position (left side) and marked it. For every procedure, all operators used the same thyroid collar with dosimeter. 

7. Are approaches to PCI predominantly femoral or radial?

Author response:

Yes, approaches to PCI are predominantly radial in this study. Radial approach has lower vascular and bleeding complications, improved patient comfort and early ambulation. We have added more details about the arterial approach in the Table 1 as follows:

Table 1 The procedural details in CA with PCI, CA, and others

8. The beam angles in Fig 1 add up to more than 100%.

Author response:

Thank you for the correction. We have modified the Figure 1:

9. One assumes that he dose-area products recorded are those for the patient. Is this correct?

Author response:

That's right. The dose area product (DAP) is obtained by DAP meters. DAP is a product of the surface area of the patient that is exposed to radiation at the skin entrance (in square centimeters or square meters) multiplied by the radiation dose at this surface (in grays). DAP is the best overall measurement of total patient dose and risks due to stochastic effects, such as DNA damage and future cancer. In clinical practice, tabular data of the conversion factors (in millisieverts per grays-centimeters squared) for DAP can yield the effective dose.

10. The fluoroscopy and acquisition times for PCI seem very short. Please double check.

Author response:

Thank you for being patient and helping me improve. We reconfirmed the fluoroscopy and acquisition times for all procedures. The acquisition time for all procedures including data conversion errors are rephrased as following:

We have rephrased the statement in line 140-141 as “The acquisition time was also longest in CA with PCI procedure (53.16 ± 10.33 s)”

Table 1 The procedural details in CA with PCI, CA, and others

11. How many procedures were used for the measurements?

Author response:

Thank you. There were 71 procedures were used for the measurements in our study. We have added more details about the number of procedures as follows:

In line 126-127 : “There were 71 procedures were included in our study, 43 CA with PCI, 16 CA and 12 others.” 

12. Was there a specific protocol for edge-to edge collimation?

Author response:

Yes, we utilized standard imaging capture protocols in the study. We used a collimation when the size of heart is greater than 2 cm in the diastolic phases to limit the images field size exposed.

13. Lines 175-176 seem to be erroneous. This study didn't report relations between beam angles and radiation dose. Was this from a previous publication.

Author response:

Thank you for the correction. We have rephrased the statement in line 174-176 as “Indeed, we analyzed the beam directions distribution complexity during the different procedures in this study.”

---

## [Decision Letter · Decision Letter 1]

22 Apr 2020

PONE-D-19-29699R1

Cardiac Catheterization Real-time Dynamic Radiation Dose Measurement to Estimate Lifetime Attributable Risk of Cancer

PLOS ONE

Dear Dr. Wu,

Thank you for submitting your manuscript to PLOS ONE. After careful consideration, we feel that it has merit but does not fully meet PLOS ONE’s publication criteria as it currently stands. Therefore, we invite you to submit a revised version of the manuscript that addresses the points raised during the review process.

Please address the minor comments from the reviewer below. 

We would appreciate receiving your revised manuscript by Jun 06 2020 11:59PM. To enhance the reproducibility of your results, we recommend that if applicable you deposit your laboratory protocols in protocols.io, where a protocol can be assigned its own identifier (DOI) such that it can be cited independently in the future. For instructions see: http://journals.plos.org/plosone/s/submission-guidelines#loc-laboratory-protocols

We look forward to receiving your revised manuscript.

Kind regards,

Jay Widmer

Academic Editor

PLOS ONE

Reviewers' comments:

Reviewer's Responses to Questions

**Comments to the Author**

1. If the authors have adequately addressed your comments raised in a previous round of review and you feel that this manuscript is now acceptable for publication, you may indicate that here to bypass the “Comments to the Author” section, enter your conflict of interest statement in the “Confidential to Editor” section, and submit your "Accept" recommendation.

Reviewer #1: All comments have been addressed

Reviewer #2: All comments have been addressed

2. Is the manuscript technically sound, and do the data support the conclusions?

Reviewer #1: Yes

Reviewer #2: Partly

3. Has the statistical analysis been performed appropriately and rigorously? 

Reviewer #1: Yes

Reviewer #2: N/A

4. Have the authors made all data underlying the findings in their manuscript fully available?

Reviewer #1: Yes

Reviewer #2: Yes

5. Is the manuscript presented in an intelligible fashion and written in standard English?

Reviewer #1: Yes

Reviewer #2: Yes

6. Review Comments to the Author

Reviewer #1: All questions that were raised have been addressed, the grammar is corrected, additional table and additional information about personal exposure has been added.

Reviewer #2: The operator exposure still needs to be more detailed.

What specific assumptions have the authors make about LAR? -a)-Constant yearly exposure from age 18 through 65? Probably not an entirely valid assumption since Cardiologist don't start at age 18, and probably don't do as many cases when starting, also have longer exposure per case when they are learning. b) -- Do operators continue through age 65 or does the LAR assume that they have stopped at the time of the study? c) is it possible to produce a survival analysis curve of the LAR?

Using a two month average is subject to determine case volume is subject to a good deal of error as there are seasonal variations in catheterization lab volume. Is it possible to use a 12 month average for the preceding year?

The quality y of Figure 1 need to be improved. It is very difficult to read the figures.

7. PLOS authors have the option to publish the peer review history of their article (what does this mean?). If published, this will include your full peer review and any attached files.

Reviewer #1: No

Reviewer #2: No

---

## [Author Response · Author response to Decision Letter 1]

26 May 2020

Dear Editor, Dear reviewers

We thank the reviewers again for the helpful comments. The following is a point-by-point response to each comment:

Reviewer #2: The operator exposure still needs to be more detailed.

1. What specific assumptions have the authors make about LAR? 

-a)-Constant yearly exposure from age 18 through 65? Probably not an entirely valid assumption since Cardiologist don't start at age 18, and probably don't do as many cases when starting, also have longer exposure per case when they are learning. 

b) -- Do operators continue through age 65 or does the LAR assume that they have stopped at the time of the study? 

c) is it possible to produce a survival analysis curve of the LAR?

Author response:

For question a and b

Thank you for pointing this out. In our study, the LAR of all cancer incidence and mortality at the radiation dose for operators were calculated by BEIR VII Report. The BEIR VII report is based on a wide enough range of exposure to support meaningful statistical modeling. However, according to the LNT model adopted in BEIR VII, 10 mSv on a yearly basis from ages 18 to 65 years old, cancer incidence was 3,059 for male and 4,295 for female. That data was directly applied to the calculation of cancer risk to medical staff, so there is limitation for generalization in our results. 

We have added more specific details in discussion as follows:

We have rephrased the statement in line 253-260 as “In addition, our study did not estimate the yearly of radiation exposure exactly and job tenure of operators. However, according to the BEIR VII report, when exposed continuously to 10 mSv on a yearly basis from ages 18 to 65 years old, cancer incidence was 3,059 for male and 4,295 for female. That data was directly applied to the calculation of cancer risk to medical staff in many studies [10, 58-60]. Although the LNT model for low-dose (<100mSv) is characterized by a great deal of uncertainty, there is no direct evidence to estimate the cancer risks for staff during the cardiac catheterization procedures.”

Newly added references:

58. Cho H-O, Park H-S, Choi H-C, Cho Y-K, Yoon H-J, Kim H-S, et al. Radiation dose and cancer risk of cardiac electrophysiology procedures. International Journal of Arrhythmia. 2015;16(1):4-10.

59. Mehta S. Health risks of low level radiation exposures: a review. Ind J Nucl Med. 2005;20:29-41.

60. Kim JB, Lee J, Park K. Radiation hazards to vascular surgeon and scrub nurse in mobile fluoroscopy equipped hybrid vascular room. Annals of surgical treatment and research. 2017;92(3):156-63.

For question c

Thank you for your kind advice. However, a survival analysis curve requires prolonged observation and studies to rule out many potentially personal, mechanical, and environmental factors. We really appreciate your precious advice and useful suggestions for our future studies.

2. Using a two month average is subject to determine case volume is subject to a good deal of error as there are seasonal variations in catheterization lab volume. Is it possible to use a 12 month average for the preceding year?

Author response:

Thank you for being patient and helping me improve. 

We have modified the case volume of 1 year occupational radiation exposure to the operators in abstract and results as follows:

We have rephrased the statement in line 32-35 (abstract) as LAR of all cancer incidences for staffs aged from 18 to 65 are varied from “0.40 %” for males to “1.50 %” for females. LAR of all cancer mortality for staffs aged from 18 to 65 are varied from “0.22 %” for males to “0.83 %” for females.

We have rephrased the statement in line 155-158 (results) as “We used a 12 month average for the preceding year to estimate the 1 year occupational radiation exposure to the operators is presented in Fig. 3. The annual average radiation dose per primary operator from all procedures was 3.49 mSv. As to the assistant operator was 1.30 mSv.”

We have rephrased the statement in line 160-162 (results) as LAR of all cancer incidence and mortality for male primary operators aged from 18 to 65 were “1.07 %”, and “0.59 %”, respectively. As to the female primary operators aged from 18 to 65 were “1.50 %”, and “0.83 %”, respectively.

We have modified the Table 2 as follows:

Table 2 Estimated LAR of all cancer incidence and mortality from cardiac catheterization procedures for operators

Operators Primary operators Assistant operators

LAR All cancer incidence aged from 18 to 65 All cancer mortality aged from 18 to 65 All cancer incidence aged from 18 to 65 All cancer mortality aged from 18 to 65

Male 1.07 % 0.59% 0.40 % 0.22 %

Female 1.50 % 0.83 % 0.56 % 0.31 %

3. The quality y of Figure 1 need to be improved. It is very difficult to read the figures.

Author response:

Thank you for the correction. We have modified the Figure 1:

---

## [Editor Report · Decision Letter 2]

28 May 2020

Cardiac Catheterization Real-time Dynamic Radiation Dose Measurement to Estimate Lifetime Attributable Risk of Cancer

PONE-D-19-29699R2

Dear Dr. Wu,

We are pleased to inform you that your manuscript has been judged scientifically suitable for publication and will be formally accepted for publication once it complies with all outstanding technical requirements.

With kind regards,

Jay Widmer

Academic Editor

PLOS ONE

---

## [Editor Report · Acceptance letter]

2 Jun 2020

PONE-D-19-29699R2 

Cardiac Catheterization Real-time Dynamic Radiation Dose Measurement to Estimate Lifetime Attributable Risk of Cancer 

Dear Dr. Wu:

I'm pleased to inform you that your manuscript has been deemed suitable for publication in PLOS ONE. Congratulations! Your manuscript is now with our production department. 

Kind regards, 

on behalf of

Dr. Jay Widmer 

Academic Editor

PLOS ONE